# An Update on Mitochondrial Reactive Oxygen Species Production

**DOI:** 10.3390/antiox9060472

**Published:** 2020-06-02

**Authors:** Ryan J. Mailloux

**Affiliations:** The School of Human Nutrition, Faculty of Agricultural and Environmental Sciences, McGill University, 21111 Lakeshore Road, Sainte-Anne-de-Bellevue, QC H9X 3V9, Canada; ryan.mailloux@mcgill.ca

**Keywords:** mitochondria, reactive oxygen species, bioenergetics, hydrogen peroxide, sex differences, substrate preferences, isopotential groups

## Abstract

Mitochondria are quantifiably the most important sources of superoxide (O_2_^●−^) and hydrogen peroxide (H_2_O_2_) in mammalian cells. The overproduction of these molecules has been studied mostly in the contexts of the pathogenesis of human diseases and aging. However, controlled bursts in mitochondrial ROS production, most notably H_2_O_2_, also plays a vital role in the transmission of cellular information. Striking a balance between utilizing H_2_O_2_ in second messaging whilst avoiding its deleterious effects requires the use of sophisticated feedback control and H_2_O_2_ degrading mechanisms. Mitochondria are enriched with H_2_O_2_ degrading enzymes to desensitize redox signals. These organelles also use a series of negative feedback loops, such as proton leaks or protein *S*-glutathionylation, to inhibit H_2_O_2_ production. Understanding how mitochondria produce ROS is also important for comprehending how these organelles use H_2_O_2_ in eustress signaling. Indeed, twelve different enzymes associated with nutrient metabolism and oxidative phosphorylation (OXPHOS) can serve as important ROS sources. This includes several flavoproteins and respiratory complexes I-III. Progress in understanding how mitochondria generate H_2_O_2_ for signaling must also account for critical physiological factors that strongly influence ROS production, such as sex differences and genetic variances in genes encoding antioxidants and proteins involved in mitochondrial bioenergetics. In the present review, I provide an updated view on how mitochondria budget cellular H_2_O_2_ production. These discussions will focus on the potential addition of two acyl-CoA dehydrogenases to the list of ROS generators and the impact of important phenotypic and physiological factors such as tissue type, mouse strain, and sex on production by these individual sites.

## 1. Introduction

The overproduction of ROS by dysfunctional mitochondria and its association with pathogenesis and ageing has been a topic of study for many years. Classically, complex I and III of the respiratory chain have been considered the sole sources of mitochondrial ROS, which, when produced in large enough quantities, damage cells, inducing disease. For example, high rates of O_2_^●−^ production during reverse electron transfer (RET) from complex II to I has been linked to ischemia-reperfusion injury to the myocardium [1]. However, although some of the deleterious effects of ROS can be experimentally reversed or prevented with mitochondria-targeted antioxidants and other factors, antioxidant therapy has been clinically shown to not be beneficial [2]. Furthermore, it has recently been demonstrated that several organisms have extended lifespans when mitochondrial ROS production is high for short periods of time [3]. Therefore, ROS can be deleterious but are also integral for cell survival and adaptive signaling. This dichotomy in the benefits and detriments of mitochondrial ROS is defined “mitohormesis”; mild oxidative stress elicits adaptive responses in cells whereas its uncontrolled overproduction can result in tissue damage [4]. To account for the positive effects of ROS on cells, new terms, specifically “oxidative eustress” and “oxidative distress”, were introduced as a direct extension from the original definition of “oxidative stress” [5]. Oxidative eustress refers to adaptive cell responses and the induction of allostasis following the reversible and site-specific protein cysteine residue oxidation/reduction, whereas the latter term relates to its production beyond the beneficial physiological range leading to cell damage and disease.

The complex relationship the mitochondrion has with ROS is underscored by evidence demonstrating that (1) mitochondria can have twelve generators that are associated with nutrient metabolism and oxidative phosphorylation (OXPHOS) [6], (2) some sites of production generate a mixture of O_2_^●−^ and H_2_O_2_ [7], and (3) mitochondria can serve as a sink for H_2_O_2_ generated by cytosolic or extracellular sources [8]. Furthermore, mitochondria invoke sophisticated mechanisms to fine tune ROS production in response to intra- and extracellular cues [9]. However, like any other second messenger, H_2_O_2_ signals need to be desensitized following a response to a physiological cue. This is achieved with antioxidant defenses, negative feedback loops that inhibit production, assembly and disassembly of supercomplexes, and proton leaks, which allows cells to use mitochondrial H_2_O_2_ for signaling whilst avoiding its deleterious effects (reviewed extensively in [10]).

Mitochondrial ROS production and homeostasis are strongly influenced by several factors. This includes substrate type, concentration, and availability as well as whether multiple substrates are supplying electrons to mitochondria, the concentration and redox state of the ROS generators, the polarity of the inner mitochondrial membrane (IMM), and accessibility of the electron donating site to molecular oxygen (O_2_) [11]. Our group also recently demonstrated that other key physiological and phenotypic factors must be considered when studying how mitochondria maintain cellular ROS balance. This includes factors such as mouse strain and sex and tissue type, which can have a strong impact on the native rate of H_2_O_2_ production by several “unconventional” ROS generators such as α-keto acid dehydrogenases [12]. Additionally, important advances have been made in identifying other ROS sources such as long-chain fatty acid dehydrogenase (LCAD) and very long-chain fatty acid dehydrogenase (VLCAD), two integral components of the fatty acid oxidation (FAO) pathway [13,14]. Therefore, the aim of the present review is to provide an update on recent advances in mitochondrial ROS production and homeostasis. 

## 2. Mitochondrial ROS Sources and Production by Reverse Electron Transfer

One of the most significant challenges facing the field of Redox Biology at this moment is decoding which enzymes in mitochondria are serving as the main source(s) of H_2_O_2_. Significant advances have been made such as the identification of S1QUELS and S3QUELS, compounds that inhibit ROS production by complexes I and III, respectively, without compromising respiration [15]. These compounds have allowed for quantitative estimation of the overall contribution of mitochondria towards cellular ROS production and have been successfully employed to demonstrate that mitochondria are the most significant H_2_O_2_ sources in cells [15]. Generally, complexes I and III are considered the only sources of mitochondrial ROS, with the latter respiratory complex serving as the main mitochondrial platform for redox signaling [16]. The caveat to this common assumption is that mitochondria can contain up to 12 ROS generators that are associated with fuel combustion and oxidative phosphorylation (Figure 1) [6,11]. Recent evidence, which has been reviewed extensively and will thus be only summarized here, has demonstrated some of these “unconventional” sources can produce more ROS than complex I and that some of these generators display native rates that are similar to complex III [11,17]. For instance, it has been demonstrated in muscle mitochondria collected from male Sprague Dawley rats that α-ketoglutarate dehydrogenase (KGDH), pyruvate dehydrogenase (PDH), and branched chain keto acid dehydrogenase (BCKDH) display higher native rates for ROS production in comparison to complex I when Krebs cycle-linked metabolites are being oxidized [18]. Similar observations were made with liver mitochondria from C57BL6N mice where KGDH and PDH accounted for up to ~45% of the ROS formed [17].

### 2.1. ROS Production by RET and Its Physiological Benefits and Pathological Consequences

Complex I is a significant ROS source during RET from complex II [1]. Traditionally, ROS genesis following RET to complex I has been associated with the pathogenesis of several diseases including heart disease (Figure 2) [1]. It was also recently shown that succinate accumulation and the subsequent induction of high rates of ROS production by RET contributes to tissue damage during organ transplantation [19]. However, short bursts in ROS production following RET from succinate can be beneficial. For instance, short bursts of ROS production by mitochondria can precondition the myocardium for protection from ischemia-reperfusion injury (Figure 2). This is achieved by the ROS-mediated induction of stress signaling and adaptive pathways that augment antioxidant defenses and protect from oxidative disstress (Figure 2) [20]. It should be noted, however, that several other studies have shown that myocardial tissue damage following ischemia-reperfusion does not occur due to RET to complex I. Indeed, it was shown recently that partial loss of complex I actually increases myocardial damage following reperfusion due to increases in ROS production by complex III [21]. Similarly, Andrienko et al. demonstrated that myocardial injury was independent of mitochondrial ROS production by RET and related to permeability transition pore opening [22]. Recent work has also demonstrated that moderate RET to complex I can increase longevity. For example, it was shown that moderate RET from complex II to I was integral for extending the lifespan of *D. melanogaster* and its adaptation to heat stress (Figure 2) [3,23]. Furthermore, disabling this pathway compromises the benefits of using RET in adaptive signaling [3,23]. It has also been shown that RET-induced ROS production by complex I following succinate accumulation drives brown fat thermogenesis (Figure 2) [24]. RET-induced ROS production by other enzymes may also play an important role in adaptive signaling. Purified KGDH and PDH produce ROS by RET from NADH [7]. It was originally suggested that RET from NADH to KGDH only occurs under pathological conditions and oxidative distress when complex I activity is compromised [25]. However, it was observed in one study that ROS production by purified KGDH and PDH can be stimulated with low amounts of NADH, which could be reproduced in permeabilized mitochondria, indicating that RET to these two enzyme complexes from NADH at its physiological concentration may also play a beneficial role in cell communication [7]. Additionally, although speculative, RET from complex II to I may also support NADH-driven ROS production by KGDH or PDH. Indeed, the successful transfer of electrons from succinate to the NAD^+^ binding site couples proton return to the production of NADH (Figure 2). This could, in turn, result in RET-driven ROS production by the α-keto acid dehydrogenases following NADH oxidation.

### 2.2. The Subcategorization of Mitochondrial ROS Producers and Identification of LCAD and VLCAD as Generators

It has been established that the twelve sites for ROS production associated with mitochondrial metabolism and aerobic respiration can be subcategorized into two groups based on the electron donating group that is involved in H_2_O_2_ generation; the NADH/NAD^+^ isopotential group and the UQH_2_/UQ isopotential group (Figure 1) [11]. In brief, the former group is comprised of flavin-dependent dehydrogenases that either reduce or oxidize nicotinamides. This includes the α-ketoacid dehydrogenases KGDH, PDH, BCKDH, and 2-oxoadipate dehydrogenase (OADH), and the flavin mononucleotide group of complex I (Figure 1) [26]. The second isopotential group contains enzymes that produce ROS by directly reducing or oxidizing the mitochondrial ubiquinone (UQ) pool. These enzymes are complex I (UQ binding site during RET), complex II, complex III, *sn*-glycerol-3-phosphate dehydrogenase (G3PDH), proline dehydrogenase (PRODH), dihydroorotate dehydrogenase (DHODH), and electron transferring-flavoprotein:ubiquinone oxidoreductase (ETFQO) (Figure 1) [26]. Although the native rates of H_2_O_2_ generation by these sites has been reviewed extensively [11,15,26], discussing this categorization is relevant here because there may be two more flavin-dependent enzymes to add to this list, namely, LCAD and VLCAD, which were recently shown to produce ROS [13,14].

Zhang and colleagues recently reported that LCAD can be a significant source of H_2_O_2_ in liver tissue [14]. Acyl-CoA dehydrogenases catalyze the first step of mitochondrial fatty acid oxidation (FAO) [14]. Mitochondria contain several of these flavoproteins which oxidize short-, medium-, long-, and very-long-chain fatty acids, (SCAD, MCAD, LCAD, and VLCAD). FAO results in the reduction of the FAD center and the passage of electrons to the UQ pool through electron-transferring flavoprotein and ETFQO. Intriguingly, H_2_O_2_ production by LCAD has often been overlooked given its absence in tissues that rely heavily on FAO to supply ATP, such as heart and muscle. However, as reported by Zhang et al., LCAD exhibits high expression in the liver and is also found in the kidney, pancreas, and lungs [14]. Coupled with this, Zhang and colleagues reported for the first time that LCAD exhibits a high rate of H_2_O_2_ generation [14]. Indeed, liver mitochondria from LCAD^−/−^ mice produce significantly less ROS and overexpression of LCAD in HepG2 cells results in a robust increase in H_2_O_2_ generation [14]. Furthermore, the purified recombinant LCAD enzyme exhibited a high rate of ROS production. Based on these findings, the authors speculated that perhaps LCAD may serve as a significant source of ROS for the pathogenesis of liver diseases. The finding that LCAD is a ROS source needs to be verified and the native rate for H_2_O_2_ production should be compared to the other established 12 sites of production to ascertain its overall contribution to mitochondrial ROS emission in the liver. This is vital since some mitochondrial ROS sources in the liver produce negligible amounts whereas PDH, KGDH, and complex III have been reported to account for ~90% of the total mitochondrial O_2_^●−^/H_2_O_2_ capacity [17].

The second enzyme that may be added to this list is VLCAD. Unlike LCAD, there has been some interest in the enzymatic properties of VLCAD since it is more broadly expressed in mammalian tissues. Kakimoto et al. investigated H_2_O_2_ generation by liver mitochondria fueled with palmitate and observed that feeding 6-week old Swiss mice a high-fat diet induced a significant increase in ROS production [13]. Additionally, the authors observed that the rate of ROS production was pH-sensitive, reaching its maximum rate of production at pH ~8.5 [13]. Based on this, the authors tested if recombinant VLCAD could serve as an ROS source, which is the acyl-CoA dehydrogenase responsible for the oxidation of C_16_–C_18_ fatty acids. Kakimoto et al. found that purified VLCAD generates H_2_O_2_ [13]. Moreover, it was observed that the rate of production was proportional to the concentration of palmitoyl-CoA [13]. These findings are in line with a study published by the same group where they were able to demonstrate that intrahepatic lipid accumulation and the induction of fatty liver disease is initiated by the overproduction of mitochondrial ROS during the oxidation of very-long-chain fatty acids following short-term feeding on a high-fat diet [27]. This resulted in the induction of oxidative damage, which preceded mitochondrial dysfunction, indicating that the pathogenesis of fatty liver disease followed by the development of steatohepatitis begins with the overproduction of ROS by VLCAD [27]. These studies were followed up by Zhang et al., where it was shown that VLCAD produces H_2_O_2_ but at a rate that was 15-fold slower than LCAD [14]. Additionally, the authors observed that liver mitochondria collected from VLCAD^−/−^ rodents did not display any difference in H_2_O_2_ production whereas a decrease was observed with samples collected from LCAD^−/−^ mice [14]. Just like LCAD, the individual native rate of ROS production by VLCAD should be compared to the other known mitochondrial ROS generators to ascertain its contribution to overall O_2_^●−^/H_2_O_2_ emission during FAO (e.g., compared to complex III and RET to complexes I and II). However, based on this preliminary evidence, it may be that the list of potential mitochondrial ROS generators may be expanded to fourteen, with LCAD and VLCAD being added to the UQH_2_/UQ isopotential group.

## 3. Sources of ROS Vary According to Tissue Type

### 3.1. ROS Production by Skeletal Muscle Mitochondria

To date, investigations into the native rates of ROS production by the 12 sites has almost been exclusively investigated using permeabilized muscle mitochondria isolated from rats. Significant observations have been made with this model system, including the demonstration KGDH and PDH can produce ~8× and ~4× more ROS, respectively, than complex I [18]. These seminal studies also identified complex II as a major contributor to mitochondrial ROS production and that OADH produces more ROS than complex I [28]. Of note as well is that the major contributors to overall ROS production are dictated by which fuels are being utilized to energize mitochondria. For example, KGDH and complexes I, II, and III are significant sources of ROS in mitochondria fueled by glutamate and malate whereas complexes I, II, and III are important generators when succinate is being oxidized [29]. Complexes I-III and ETFQO are major generators in isolated rat muscle mitochondria incubated in buffers that mimic physiological substrate availability when muscles are at rest [29]. By contrast, only complex I serves as the major source when incubation conditions are set to mimic intense exercise [29]. In addition, the substrate type and combination dictate the rate of overall ROS production by permeabilized muscle mitochondria. For example, incubating muscle mitochondria in succinate induces the highest rates of ROS production, whereas fueling samples with glutamate and malate or palmitoyl-CoA results in lower rate of generation [29]. Overall, these seminal findings demonstrate that mitochondria can adopt dynamic “ROS release signatures” that can be strongly influenced by physiological changes in substrate type and availability.

### 3.2. ROS Production by Liver Mitochondria

Of note regarding substrate type and availability is that different tissues display substrate preferences in response to dietary changes and shifts in physiological demands (e.g., hepatocytes switch to burning fatty acids as a fuel during starvation to maintain blood glucose levels using gluconeogenesis). Thus, it is likely that mitochondria in different tissues rely on a select set of the 12 potential ROS sources to serve as the main generators of cellular H_2_O_2_. In 2016, it was reported that both purified KGDH and PDH of porcine heart origin can generate H_2_O_2_ by forward and reverse electron flow [7]. This led to the development of the hypothesis that KGDH and PDH may also serve as important ROS sources during ischemia-reperfusion injury to the myocardium. Intriguingly, it was observed that both KGDH and PDH displayed low rates of ROS production in cardiac mitochondria, whereas complexes I and III were identified as the major H_2_O_2_ sources [21,30]. Further investigations revealed that KGDH and PDH produce more ROS in liver mitochondria when compared to samples collected from cardiac tissue, even though there was no difference in the expression levels of either enzyme complex in both tissues [7]. A follow-up study by the same group in 2017 demonstrated that KGDH and PDH account for ~45% of the total ROS produced by liver mitochondria, whereas both enzymes exhibit low native rates of production in cardiac tissue (Figure 3) [17]. Additionally, liver mitochondria energized with α-ketoglutarate or pyruvate exhibit rates of H_2_O_2_ production that are ~7-fold higher than cardiac mitochondria [30]. These observations are supported by a study that employed a transgenic mouse model for glutaredoxin-2 (GRX2), a thiol oxidoreductase that catalyzes the reversible inhibition of mitochondrial enzymes by protein *S*-glutathionylation. GRX2 is required to *S*-glutathionylate target proteins in response to glutathione pool oxidation, which inhibits ROS production [30]. Once the redox potential of the glutathione pool is restored, GRX2 catalyzes the deglutathionylation of these targets. ROS production is significantly decreased in liver mitochondria collected from mice containing a deletion for GRX2 [30]. This is associated with the increased glutathionylation of KGDH and PDH and decreased ROS production by both enzymes [30]. Intriguingly, the opposite trend was observed in cardiac tissue. Loss of GRX2 increased ROS production [30]. Furthermore, this increase was attributed to the glutathionylation of the respiratory complexes [30]. What is noteworthy, however, is like muscle, liver mitochondria can dynamically change ROS sources in response to the type of substrate being oxidized. For instance, complexes II and III account for ~95% of the ROS produced by liver mitochondria when choline and dimethylglycine, substrates that donate electrons directly to the UQ pool, are being oxidized (Figure 3) [31]. Similarly, complexes II and III are the only producers when succinate serves as a substrate. Another surprising feature of mitochondrial ROS production by liver mitochondria oxidizing succinate, choline, or dimethylglycine is that complex I serves as a negligible source [7,31]. Thus, in aggregate, liver mitochondria can also adopt dynamic ROS signatures depending on which substrate is being oxidized with complex III serving as a main source and complex I generating negligible amounts regardless of which fuel is being metabolized.

### 3.3. ROS Generation by Cardiac Mitochondria

Cardiac mitochondria almost exclusively produce ROS by complexes I, II, and III. Cardiac mitochondria oxidizing succinate alone displayed rates of H_2_O_2_ generation that were almost 4-fold higher when compared to liver samples isolated from the same mouse (Figure 3) [30]. Loss of GRX2 results in a two-fold increase in ROS production in mitochondria oxidizing succinate, α-ketoglutarate, or pyruvate [30]. Additionally, complexes I and III were identified as the major ROS sources in cardiac mitochondria energized with succinate, α-ketoglutarate, or pyruvate, with KGDH and PDH making small contributions to overall production [30]. Therefore, in contrast to liver and muscle, heart mitochondria rely on complexes I and III as its main ROS suppliers. As noted above, both tissues display no differences in KGDH and PDH expression nor is there any significant alteration in the levels of respiratory complex subunits. Although speculative at this point, differences in which sites serve as the highest ROS generators may be related to the bioenergetics of these mitochondria. Indeed, the rate of respiration is lower in liver mitochondria, which may be related to lower NADH turnover by complex I. State 3 respiration in cardiac mitochondria oxidizing either pyruvate or lactate with malate is ~5-fold higher when compared to liver mitochondria [32]. Similar observations were made with skeletal muscle mitochondria, where it was ~3-fold higher [32]. The lower rate of NADH turnover would result in the allosteric inhibition of PDH and KGDH and the subsequent increase in ROS production. Furthermore, as noted above, PDH and KGDH can produce ROS by RET from NADH. These differences in bioenergetics may be related to the degree of MIM folding and cristae formation [33]. Much more work is required to fully catalogue the tissue-specific differences in the ROS release rate of the 12 accepted individual sites of production. The study described above is not the first of its kind to demonstrate that mitochondria isolated from different tissues display variances in fuel preference.

## 4. Impact of Rodent Strain on the Individual Sites of ROS Production

Several rodent strains are utilized to advance our collective understanding of how mitochondria use ROS in adaptive signaling and the contribution of defects in redox signaling towards the induction of diseases. The C57BL6N (6N) and C57BL6J (6J) are the most popular mouse strains used for human disease modeling, in particular when studying relationships between the overproduction of ROS by mitochondria in various tissues and the development of diet-induced obesity and obesity-related disorders (e.g., fatty liver disease and insulin resistance), heart disease, and myocardial damage following ischemia-reperfusion. What is rarely considered in these studies, however, is whether there are strain-dependent effects on the rate of H_2_O_2_ production by the twelve individual generators associated with mitochondrial energy metabolism. Although strain effects on cellular redox buffering capacity have been investigated [34], only one study has directly compared the H_2_O_2_ forming capacity of some of the twelve sites of production. This study was done using permeabilized muscles from mice, rats, and humans [35]. In this case, it was shown that the rate of production by PDH in all three species depends on GSH availability [35]. These comparisons are vital for translational medicine since the goal is to utilize rodents to accurately model human diseases. Second, several rodent strains carry loss-of-function mutations in proteins that are integral for maintaining cell redox balance. This can affect ROS release profiles of the individual sites through alterations in redox buffering capacities. For instance, loss-of-function variance in NNT, a mitochondrial inner membrane protein vital for the provision of NADPH, has been shown to pre-dispose 6J mice to the development of glucose intolerance, insulin resistance, diet-induced obesity, and glucocorticoid deficiency [26]. Recent research efforts attempted to discern if development of these diseases was attributed to the overproduction of ROS by mitochondria followed by the induction of oxidative stress. Intriguingly, Fisher-Wellman et al., observed that there were no significant differences in the overall ROS production by muscle mitochondria isolated from 6J and 6N mice fed a high-fat diet [36]. However, in a separate study, the same group found 6J muscle mitochondria produced more H_2_O_2_ in comparison to 6N mice but only when pyruvate served as the substrate, indicating that PDH may serve as a more potent ROS generator in the muscles of this mouse strain [37]. Similarly, our group reported on several occasions that KGDH and PDH are good ROS generators in liver mitochondria isolated from 6N mice [17,30]. However, this was not the case in B6.129S4 mice where KGDH and PDH made minor contributions to the overall native rate of ROS release by liver mitochondria [21]. Collectively, these preliminary findings indicate that mouse strain strongly affects which of the twelve sites of production will serve as a major ROS producer.

### 4.1. ROS Sources in Liver Tissue Vary According to Mouse Strain

Our group recently profiled the native rates of ROS production by several ‘unconventional’ generators using mitochondria collected from 6N and 6J mice. This included measuring the rates of production by KGDH, PDH, BCKDH, complex II, PRODH, and G3PDH in liver mitochondria and KGDH, PDH, PRODH, and G3PDH in heart mitochondria, respectively [12]. Mitochondria were permeabilized with Triton-X and treated with complex I and III inhibitors to curtail ROS production by the electron transport chain. Additionally, complex II inhibitors were included when PRODH and G3PDH measurements were conducted. Unsurprisingly, it was found that PDH and KGDH exhibited high rates of H_2_O_2_ production in liver mitochondria collected from 6N mice [12]. Complex II was also an important site of production, whereas BCKDH, PRODH, and G3PDH generated negligible amounts [12]. Additionally, measurements conducted with intact mitochondria confirmed that PDH, KGDH, and complex II were important ROS generators; however, BCKDH, PRODH, and G3PDH displayed higher rates of production, suggesting that H_2_O_2_ production by these sites is influenced by the mitochondrial membrane potential. By contrast, liver mitochondria collected from 6J mice displayed a completely different ROS production profile. All three α-keto acid dehydrogenases were found to display modest to high rates of H_2_O_2_ production in permeabilized and intact liver mitochondria [12]. Additionally, all three UQH_2_/UQ isopotential enzymes measured also displayed modest rates of production. It was also found that overall ROS production was significantly lower in 6J liver mitochondria, an effect that was attributed to a compensatory increase in redox buffering capacity due to higher mitochondrial catalase and glutathione peroxidase levels [12]. Furthermore, 6J mitochondria produced significantly less ROS when operating under state 3 or 4 respiratory conditions. Overall, these findings indicate that 6J mice invoke several adaptations to compensate for the loss of NNT, which includes restructuring metabolic pathways and increasing the expression of catalase and glutathione peroxidase.

### 4.2. ROS Sources in Cardiac Cells are Affected by Mouse Strain

Cardiac mitochondria from 6N and 6J mice also displayed some fundamental differences in the ROS-producing potential of several individual sites of generation. PDH and KGDH exhibited low native rates of production, confirming previous observations made by our group [12]. G3PDH displayed high rates of production in mitochondria collected from both mouse strains [12]. Intriguingly, PRODH also displayed a high rate of production but only in 6J mitochondria. Furthermore, these trends for the individual rates of ROS production by these sites were also observed in intact mitochondria. Measurement of overall ROS production by mitochondria energized with proline, pyruvate, α-ketoglutarate, and glycerol-3-phosphate revealed that 6J heart samples produced more ROS, which was attributed to inefficient electron transfer through the respiratory chain [12]. PRODH displays low expression in many rodent tissues and is thus reported as a low ROS emitter [38]. Certain organisms, such as flight insects, express high amounts of PRODH since proline is the primary fuel utilized for wing function [38,39]. However, this also means PRODH is a major ROS producer in these organisms [39]. In rodents and humans, increased PRODH expression is often associated with the pathogenesis of several disorders including cancer. Increased PRODH expression has been linked to the induction of oxidative distress, DNA damage, senescence, and apoptosis in mammalian cells, which is related to the overproduction of ROS [40]. Moreover, characterization of the responses to acute myocardial infarction in the Collaborative Cross revealed that 6J mice displayed the highest mortality rate among the eight inbred strains investigated, which is likely due to increased ROS production and diminished protection from oxidative stress due to the absence of NNT [41]. As noted in our study, it was found that PRODH displays a ~2-fold increase in H_2_O_2_ production, which could also contribute to the induction of oxidative distress and the induction of myocardial tissue damage [12]. Overall, this underscores the need for strongly considering strain-differences when evaluating the relationships between mitochondrial ROS production and cell signaling using rodent models for human diseases.

## 5. Sex Differences in Mitochondrial ROS Production

Sex differences in the physiological and biochemical traits are common in mammals and humans and result from variances in genetic and epigenetic factors and cell signaling and communication. The significant effect of sex on physiology is underscored by its impact on lifespan and the manifestation of several disorders [42]. For instance, pre-menopausal women display lower rates of metabolic syndrome and development of kidney disease when compared to men, an effect that is lost with the sharp decline in 17-β-estradiol production by granulosa cells in ovarian follicles [42,43]. Investigating sex differences in redox buffering capacity at the cellular and mitochondrial level has become a major focus in understanding the impact of sex on the manifestation and progression of many disorders. This should not be surprising since the cell redox buffering system serves as a critical interface for dictating changes in cell behavior in response to environmental cues (e.g., hormonal signaling) [44]. Sex has a strong impact on cell redox buffering capacity. Mitochondria from the livers of female rodents display higher rates of state 3 respiration, cardiolipin content, and increased oxidative phosphorylation following consumption of a high-fat diet [42]. Additionally, these mitochondria produce less ROS and contain higher levels of antioxidant defenses [39]. Similarly, brain and heart mitochondria from female rodents generate less ROS and have a higher redox buffering capacity [24]. 

Interest in understanding the effect of sex on cell redox buffering capacity was recently extended to interrogating the impact of 17-β-estradiol availability on mitochondrial ROS production and bioenergetics and the relationship of these parameters to the development of metabolic syndrome. A recent study found that surgical removal of the ovaries in female mice (ovariectomy (OVX); menopause-like symptoms model) results in a significant increase in mitochondrial ROS production due to oxidation of mitochondrial redox buffering networks [45]. Interestingly, these effects can be reversed by treating OVX mice with 17-β-estradiol [45]. The positive effects of 17-β-estradiol on mitochondrial ROS production, bioenergetics, and redox balance were attributed to its incorporation into the MIM, which improved mitochondrial membrane fluidity [45]. However, it is also documented that 17-β-estradiol can induce cell signals that augment the expression of genes involved in antioxidant defense and mitochondrial proliferation and bioenergetics [46]. 

To date, only one study has focused on sex differences in the production of ROS by the 12 individual sites in mitochondria. Although these concepts are still preliminary at best, it has revealed some intriguing sex differences regarding mitochondrial ROS production by PDH, KGDH, and the respiratory chain, respectively [47]. One intriguing difference was the rate of ROS production by PDH and KGDH in liver mitochondria. PDH and KGDH displayed a ~3.5-fold and ~5-fold higher rate of H_2_O_2_ production in male liver mitochondria when compared to female littermates [47]. By contrast, succinate-stimulated ROS production was significantly higher in female liver mitochondria. Interestingly, the authors simultaneously identified the first sex dimorphism in mitochondrial redox signaling through protein *S*-glutathionylation [47]. Indeed, *S*-glutathionylation has been identified as an integral negative feedback loop for the regulation of mitochondrial ROS production by PDH, KGDH, and complexes I and II [48]. Deletion of the *Grx2* gene results in the increased glutathionylation of PDH and KGDH, which inhibits ROS production in mitochondria collected from male mice [47]. However, in comparison to male mice, deleting the *Grx2* had no effect on ROS release by PDH, KGDH, and the respiratory chain in liver mitochondria from female littermates. This indicates that female liver mitochondria are not reliant on glutathionylation to negatively regulate ROS production, which may be attributed to its higher redox buffering capacity [47]. In muscles, surprisingly, PDH, KGDH, and the respiratory chain produced more ROS in female mitochondria (~3-fold higher for PDH and KGDH and ~5-fold higher for the respiratory chain) [47]. However, despite this difference, female muscle mitochondria did not require *S*-glutathionylation for the negative regulation of ROS production, further demonstrating that sex strongly influences cell redox state. 

## 6. Conclusions

Emergence of novel approaches and sensitive intracellular techniques, such as protein-based probes that detect H_2_O_2_ and novel ROS production inhibitors that do not interfere with mitochondrial respiration, have established that ROS are integral second messengers for cell signaling. This communication can occur directly through the reversible oxidation of protein cysteine thiols or indirectly via the oxidation of cell redox buffers, such as glutathione, resulting in the glutathionylation of proteins. The balanced use of H_2_O_2_ and redox buffers in allostatic eustress signaling has been called the “Golden Mean of Healthy Living”, which is required to achieve new levels of adaptative homeostasis in response to the sum of the environmental and physiological exposures experienced by cells [49]. Therefore, decoding how mitochondria produce H_2_O_2_ is of vital importance for furthering our understanding as to how mammalian cells achieve this “Golden Mean”. In this updated review, I present new insights into the complexities and challenges associated with understanding mitochondrial H_2_O_2_ production. This includes a suggestion of adding LCAD and VLCAD to the growing list of potential mitochondrial ROS generators and considering which individual sites serve as major contributors towards overall production in different tissues. Of the utmost importance is the dearth of information available on the impact of sex- and tissue-dependent substrate preferences on the native rate of ROS production by the different enzymes in the NADH/NAD^+^ and UQH_2_/UQ isopotential groups. This is of vital importance given that these factors have a profound effect on mitochondrial and cellular redox buffering capacities and ROS generation. Overall, tremendous strides have been made in solidifying our collective understanding of how mitochondria utilize H_2_O_2_ in second messaging. However, as presented here, more work is required to fully understand tissue-dependent substrate preference and changes in fuel supply on ROS production by the individual sites and the effect of sex and hormonal dimorphisms on rates of ROS generation.

## Figures and Tables

**Figure 1 antioxidants-09-00472-f001:**
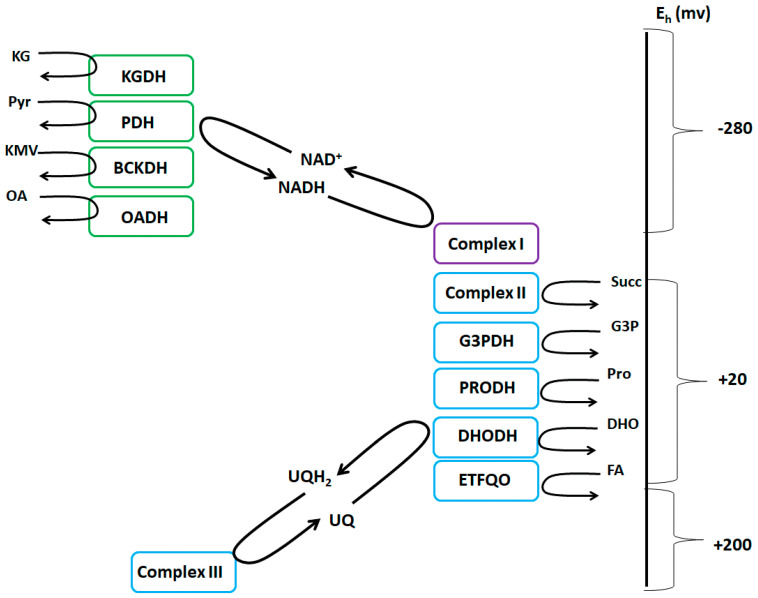
The NADH/NAD^+^ and UQH_2_/UQ isopotential groups. The NADH/NAD^+^ group (green) consists of KGDH (α-ketoglutarate dehydrogenase), PDH (pyruvate dehydrogenase), BCKDH (branched-chain keto acid dehydrogenase), OADH (2-oxoadipate dehydrogenase), and complex I. The UQH_2_/UQ isopotential group (blue) is made up of complex II, G3PDH (*sn*-glycerol-3-phosphate dehydrogenase), PRODH (proline dehydrogenase), DHODH (dihydroorotate dehydrogenase), ETFQO (electron transferring-flavoprotein:ubiquinone oxidoreductase), and complex III. Complex I is denoted in purple since it uses both isopotential groups to form ROS and can produce ROS from both its flavin mononucleotide group and ubiquinone binding site, respectively.

**Figure 2 antioxidants-09-00472-f002:**
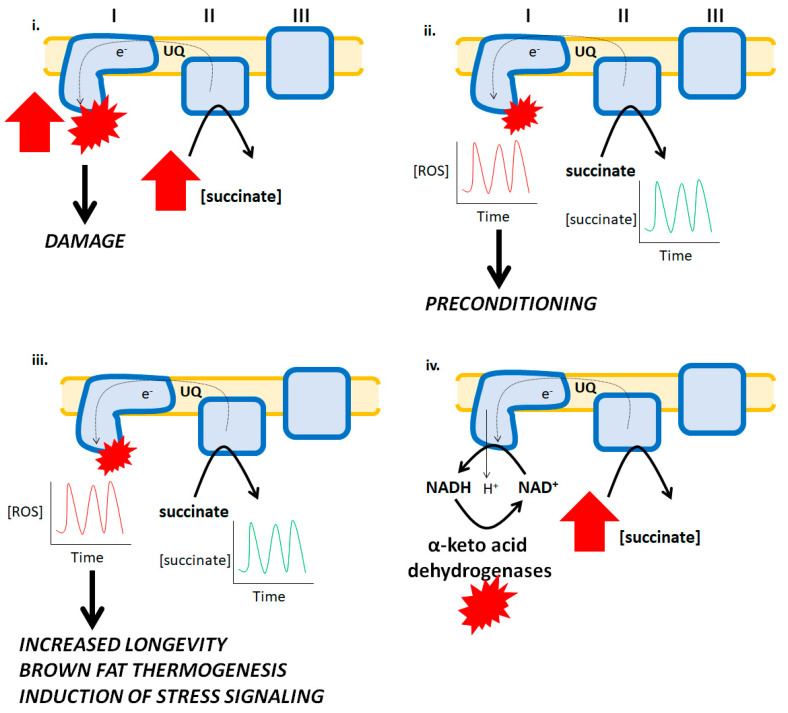
Reverse electron transfer (RET) and its pathological effects and physiological benefits. **i**. The accumulation of succinate during ischemia results in a high burst in ROS production by complex I. Succinate is oxidized by complex II and due to the hyperpolarization of the mitochondrial inner membrane and over-reduction of UQ pools, electrons flow backwards to complex I forming high amounts of ROS culminating with myocardial tissue damage and the development of heart disease. **ii.** Ischemic preconditioning is associated with temporary bursts in ROS production by mitochondria which activates cellular programs involved in adaptative signaling resulting in a bolstering of antioxidant systems that protect cardiomyocytes from oxidative stress. In this hypothetical scenario, the short bursts in ROS production are caused by spatio-temporal increases and decreases in succinate, which results in short and temporary bursts in ROS production by RET. **iii.** Increases and decreases in succinate availability are integral for driving RET-mediated ROS production in brown fat and other tissues, contributing to vital physiological functions such as thermogenesis, stress signaling, and longevity. **iv.** A hypothetical mechanism for how RET from succinate to complex I drives ROS production by α-keto acid dehydrogenases. Reverse electron flow drives proton return to the matrix by complex I resulting in NADH formation and its subsequent oxidation by the dehydrogenases. This leads to ROS production by these enzymes.

**Figure 3 antioxidants-09-00472-f003:**
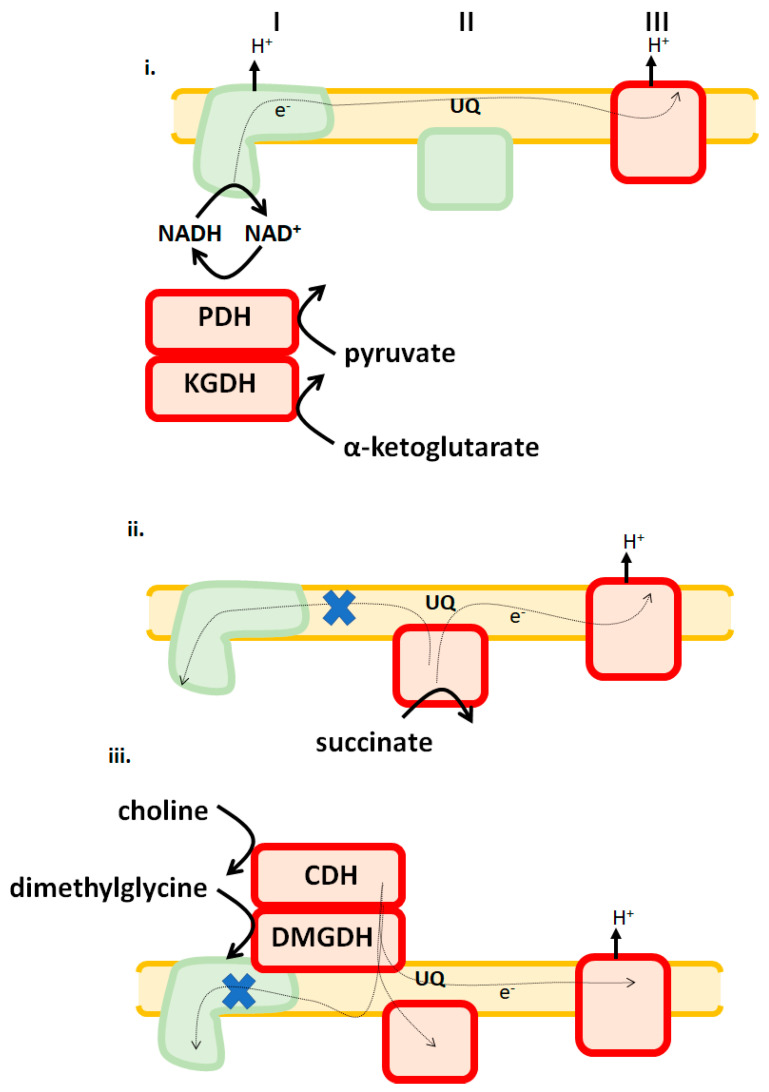
Liver mitochondria adopt different ROS release signatures in response to which substrates are being used as fuels. **i**. Pyruvate dehydrogenase (PDH), α-ketoglutarate dehydrogenase (KGDH), and complex III are major sources of ROS when Krebs cycle-linked metabolites are being oxidized. **ii.** Complexes II and III are major sources when succinate feeds electrons directly into the UQ pool are being oxidized. **iii**. Both choline and dimethylglycine, which are oxidized by choline dehydrogenase (CDH) and dimethylglycine dehydrogenase (DMGDH), generate ROS by RET to complex II and forward flow to complex III. X indicates RET is blocked. Enzymes denoted in red are significant ROS sources. Green indicates that it is a negligible source under those bioenergetic conditions.

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
