# Peer review of "An Update on Mitochondrial Reactive Oxygen Species Production"

_antioxidants, 2020, doi:10.3390/antiox9060472_

Round 1

Reviewer 1 Report

This review is quite good review for mitochondrial ROS and those function.

Introduction

Line 40; 40 years is too long.

Please see this paper.

Indo HP, Hawkins CL, Nakanishi I, Matsumoto K, Matsui H, Suenaga S, Davies MJ, St Clair DK, Ozawa T, Majima HJ. Role of Mitochondrial Reactive Oxygen Species in the Activation of Cellular Signals, Molecules and Function, Handb Exp Pharmacol. (HEP) Vol 240, Pharmacology of Mitochondria pp 439-456, 2017 Feb 8. doi: 10.1007/164_2016_117

Abstract

Mitochondria are a major source of intracellular energy and reactive oxygen species in cells, but are also increasingly being recognized as a controller of cell death. Here, we review evidence of signal transduction control by mitochondrial superoxide generation via the nuclear factor-κB (NF-κB) and GATA signaling pathways. We have also reviewed the effects of ROS on the activation of MMP and HIF. There is significant evidence to support the hypothesis that mitochondrial superoxide can initiate signaling pathways following transport into the cytosol. In this study, we provide evidence of GATA signal transductions by mitochondrial superoxide. Oxidative phosphorylation via the electron transfer chain, glycolysis, and generation of superoxide from mitochondria could be important factors in regulating signal transduction, cellular homeostasis, and cell death.

Author Response

1)The introduction has been revised and shortened

2) This line has been corrected.

3) I have added this citation in the introduction.

Reviewer 2 Report

Review of ‘’An update on mitochondrial reactive oxygen species production’’

The mitochondrion generates H2O2, which is able to move to the other cellular compartments and serves as a signaling agent at normal conditions, or damages macromolecules (lipids, proteins & DNA) at higher levels.  The topic of mitochondrial oxidative stress is not only very interesting but quite relevant for understanding the development of many pathologies. Updating and analyzing this issue is obviously important provided that the present review adds substantially to the existing literature. However, as a whole, the manuscript is not well written, and there are major concerns that need to be addressed.

The Introduction section is unnecessarily long, not focused, and does not introduce in a practical way the rationale and the precise objectives of the review. It is in a new section (or subsequent background) that several points or aspects of the Introduction can be developed harmoniously. The section ‘’Expanding list of mitochondrial ROS sources’’ is not constructed clearly and smoothly so that readers have a quick opportunity to understand the different sources of ROS. It is rather confusing when it could be direct and supported by a more adequate illustration or table detailing the sources of ROS, which are anyway reported by various articles. Why not present conventional and unconventional mitochondrial sources of ROS in an illustration/table and discuss them appropriately and critically in the text? ‘’Tissue type and its effect on mitochondrial ROS production’’: This section is ambiguous and not properly structured. Based on the section title and what was mentioned in the objectives, we expect to see the ROS production in several tissues and organs, but only the liver and the heart are developed. Key comparative points among various organs in a summarizing table would provide a picture of what constitutes a pluralistic analysis.  ‘’Impact of rodent strain on the individual sites of ROS production’’: A detailed comparative table would facilitate the comprehension and understanding of readers. Furthermore, skipping from topic to topic does not provide reader-friendly and simple messages.  The same comments also apply to the section of ‘’Sex differences in mitochondrial ROS production’’. Typographical errors and syntax problems are often noted along the text and lessen even more the quality of the manuscript.

Author Response

I have made extensive revisions to the manuscript.

Reviewer 3 Report

This is a litany of the contributions of various mitochondrial enzymes to superoxide and hydrogen peroxide production and effects of gender and strain of animal models. 

  1. While reviewing the conditions under which ROS are generated is of interest, it would be helpful to extract some general principles or some key questions from this. Why does it matter which enzymes are producing ROS?  Do those conditions (metabolic pathways) under which more ROS are produced correlate with an upregulation of antioxidant defenses?  It would be helpful to have some general hypotheses to help frame the discussion about what conditions, strains, gender lead to more ROS production and which lead to less.
  2. The line “mitochondria can contain up to sixteen ROS sources, twelve of which are associated with nutrient metabolism and oxidative phosphorylation (OXPHOS)” is repeated in the Abstract (l. 15-16), Introduction (l. 51-52) and Section 2 (l. 86-87). It is also stated that “the list of potential mitochondrial ROS generators may be expanded to eighteen, with LCAD and VLCAD being added (l. 171-172). This gives the impression that formulating this list is important, yet the figure shows only 11 enzymes.  The missing five enzymes and the reason they are not mentioned should be added.  Their omission suggests that they do not make a significant contribution to ROS generation and should be dropped from the list.
  3. The last line of the abstract states “Finally, I will address the overlooked importance of mitochondrial catalase in the maintenance of the cellular H2O2 budget.” I did not see anything substantial about catalase in the review, and that line of the abstract should be deleted.

The manuscript contains a number of lesser errors which require attention:

  1. 75: Should “to” be “two”?
  2. 95: “formed produced” one word should be deleted
  3. 135: I would suggest a new paragraph beginning with “Zhang and colleagues”
  4. 197: Should “hear” be ”heart”?
  5. 201: “where” should be “were”
  6. 243: I would suggest inserting the word “mouse” between “popular” and “strains”
  7. 247: “the” should be deleted before “whether”
  8. 248: insert “production” after “H2O2
  9. 373: insert “of” after “reversible oxidation”

Author Response

1) This is a good point. The induction of antioxidant defense systems in response to differences in ROS production by these individual sites has, unfortunately, not been investigated in detail. These limitations have been reviewed previously and were thus not discussed here. Instead, the aim of the present review was to demonstrate that important factors such as genetic and sexual variances and tissue type have a strong impact on rates for ROS production by mitochondria and which sources generate the most.

2) This has been clarified since complex I can generate ROS from both its FMN and UQ binding sites. Additionally, emphasis was placed only on the 12 sites involved in nutrient metabolism. The others (e.g. monoamine oxidase) were excluded since the aim of the review was to discuss new developments in ROS production by these 12 sites.

3) Removed.

Reviewer 4 Report

This review article describes mitochondrial ROS production sites and their relative contributions in different tissues and substrates. The author also discussed differences of rodent strains and sex in individual ROS sites. This is a topic of interest and will provide good information. Following improvements are recommended.

  1. In the abstract, line 19, antioxidants may not be the only one. May add others such as proteins regulating metabolism, OXPHOS, and redox balance.
  2. The last sentence of the abstract indicates that the review will discuss the importance of mitochondrial catalase. However, it barely describes mitochondrial catalase. However, the question is whether catalase is actually present in mitochondria in mammals.
  3. Line 86-87: need to add what the other 4 sites are.
  4. Line 123- 133 needs to be moved up right after the “(Figure 1)[8, 12]” in line 87.
  5. Line 135 LCAD and VLCAD: with the way it was written, it is not clear whether LCAD or VLCAD itself can produce ROS. LCAD and VLCAD are the acyl-CoA dehydrogenase that reduces FAD and its electrons are transferred to the CoQ via ETFQO. This means ETFQO is the site of the ROS generation, not the LCAD and VLCAD. Probably, the author may need to explain the experiments with the recombinant LCAD or VLCAD generating ROS in a more detailed manner, which would be the more direct evidence that enzymatic reaction of VLCAD or LCAD itself can produce ROS.
  6. Line 197: change “hear” to “heart”
  7. Line 201: change “where” to “were”
  8. Line 236: the sentence needs references.
  9. Section 4 (rodent strain difference) may be a good place to discuss differential expression of SCAF1 (supercomplex assembly factor 1) in different mouse strains (Science 2013, vol 340: 1567). As this changes substrate-dependent respiration rates, it may affect ROS production as well.
  10. Line 363: Add “in female mice” after “in liver mitochondria”
  11. Line 373: “…reversible oxidation protein…”. Add “of”
  12. Line 374: “…via oxidation cell redox …”. Add “of”

Overall, better organization may be necessary. In current form, sections are quite long and the readers may lose what they read. Adding subheadings and describing narrow focused subject under each subheading will help reading and understanding the article.

Author Response

  1. Corrected.
  2. Corrected
  3. Corrected
  4. Corrected
  5. Corrected
  6. Corrected
  7. Corrected
  8. Corrected
  9. The objective of the review was to discuss new developments in ROS production by the 12 sites.
  10. Corrected
  11. Corrected
  12. Corrected.
  13. Subheadings have been added

Round 2

Reviewer 2 Report

NA

Author Response

I have been through the manuscript extensively and corrected any spelling errors.

Reviewer 3 Report

The addition of two illustrations and additional text are a considerable improvement to the readability of the manuscript.  I appreciate the author's effort in making these adjustments.

The final line in the abstract relating to mitochondrial catalase has not been removed and should be.  This topic is not included in the review.

Additional typographical corrections:

l. 21: "understand" should be "understanding"

l. 54: "as well" should be "as well as"

l. 79:  "generates" seems to be out of place and unnecessary

l. 107:  insert "in" between "increases" and "ROS"

l. 110:  insert" and" between "beneficial" and "can"

Figure 2 legend:  "short temporary" is a tautology.  Would suggest "short repeated" or "short intermittent" instead

l. 226:  "are being oxidized" should be deleted

Author Response

This line has been removed.

1) corrected.

2) corrected.

3) corrected.

4) corrected.

5) corrected.

6) corrected.

7) corrected.